# The Effects of Aging on Rod Bipolar Cell Ribbon Synapses

**DOI:** 10.3390/cells12192385

**Published:** 2023-09-29

**Authors:** Abhishek P. Shrestha, Nirujan Rameshkumar, Johane M. Boff, Rhea Rajmanna, Thadshayini Chandrasegaran, Courtney E. Frederick, David Zenisek, Thirumalini Vaithianathan

**Affiliations:** 1Department of Pharmacology, Addiction Science, and Toxicology, College of Medicine, University of Tennessee Health Science Center, Memphis, TN 38163, USA; 2Department of Zoology, Faculty of Science, University of Jaffna, Jaffna 40000, Sri Lanka; 3Department of Molecular and Cellular Physiology, Yale University School of Medicine, New Haven, CT 06510, USAdavid.zenisek@yale.edu (D.Z.); 4Department of Ophthalmology, Hamilton Eye Institute, College of Medicine, University of Tennessee Health Science Center, Memphis, TN 38163, USA

**Keywords:** aging, retina, vision, ribbon synapses, electrophysiology, calcium dynamics, confocal imaging, patch clamping

## Abstract

The global health concern posed by age-related visual impairment highlights the need for further research focused on the visual changes that occur during the process of aging. To date, multiple sensory alterations related to aging have been identified, including morphological and functional changes in inner hair cochlear cells, photoreceptors, and retinal ganglion cells. While some age-related morphological changes are known to occur in rod bipolar cells in the retina, their effects on these cells and on their connection to other cells via ribbon synapses remain elusive. To investigate the effects of aging on rod bipolar cells and their ribbon synapses, we compared synaptic calcium currents, calcium dynamics, and exocytosis in zebrafish (*Danio rerio*) that were middle-aged (MA,18 months) or old-aged (OA, 36 months). The bipolar cell terminal in OA zebrafish exhibited a two-fold reduction in number of synaptic ribbons, an increased ribbon length, and a decrease in local Ca^2+^ signals at the tested ribbon location, with little change in the overall magnitude of the calcium current or exocytosis in response to brief pulses. Staining of the synaptic ribbons with antibodies specific for PKCa revealed shortening of the inner nuclear and plexiform layers (INL and IPL). These findings shed light on age-related changes in the retina that are related to synaptic ribbons and calcium signals.

## 1. Introduction

Visual impairment represents a significant global health concern that affects millions of people. The underlying causes of aging are intricate and involve factors such as abnormal mitochondria, epigenetic alterations, elevated levels of reactive oxygen species (ROS), and a reduction in the length of chromosomal telomeres [1,2,3]. Age-related changes are particularly evident in the gradual loss of sensory systems such as hearing and vision [4]. In the auditory system, age-related hearing loss is linked to the decline of inner hair cell ribbon synapses and decreased hearing sensitivity, leading to decreased speech comprehension [5]. Hallmark changes to the visual system that specifically affect the retina include neuronal loss in the macula, tissue thinning, increased retinal pigment epithelium, and reduced visual function [3,6,7,8].

Of note, the decline in scotopic vision is much more evident than in cone-mediated photopic vision, indicating that the rod pathway is more susceptible to the effects of aging than cone-mediated vision [9]. Rod-generated signals pass to the inner retina via the rod bipolar cell, and in diurnal Chilean Degu (*Octodon degus*), more significant age-related degeneration was observed in a number of rod bipolar cells, dendrites, and terminals than in those of younger rodents [10]. Despite the extensive characterization of age-related changes in the retina and sensory systems, functional changes in rod bipolar cells and their ribbon synapses have not been deeply explored.

To better define the impact of aging on vision, we investigated the changes in the function and morphology of rod bipolar cell ribbon synapses in zebrafish (*Danio rerio)*. Zebrafish are highly effective model organisms because they are easily maintained and cost-effective, attain sexual maturity after 3–4 months, and yield 200–300 offspring on a weekly basis [11]. Most importantly, when the protein-coding genes of zebrafish and humans are compared, 71% of human genes have at least one orthologue in the zebrafish genome, with 82% of human disease-related genes exhibiting homology with at least one zebrafish gene [12]. The short life span of zebrafish provides a unique advantage to studying the progression of aging using markers comparable to those used in humans. For example, aged zebrafish often display spinal curvature, cognitive impairment, and visual impairments such as cataracts [13,14].

The primary aim of the current study was to investigate age-related changes in the function and morphology of rod bipolar cell ribbon synapses and synaptic ribbons from zebrafish retinas. Our approach combined patch-clamp electrophysiology, high-resolution calcium imaging, and semi-quantitative immunohistochemistry in the retinas of zebrafish of the wild Indian karyotype (WIK) that were categorized as middle-aged (MA, 18-months-old) or older-aged (OA, 36-months-old). These age groups were selected to align with those used in previous studies that demonstrated detrimental sub-organismal effects of aging in zebrafish [15,16,17,18,19]. Based on previously reported age correspondence, these age groups correspond to human ages of approximately 38 and 75 years of age, respectively [20]. Bipolar ribbon synapses can transmit release signals for both tonic and phasic signals in response to sustained depolarization [21,22,23,24,25,26,27,28,29,30,31]. Here, we focused primarily on comparing the release properties to brief stimuli, local calcium signals, and synaptic ribbons in the MA and OA zebrafish.

## 2. Materials and Methods

### 2.1. Zebrafish Rearing

Male and female middle-aged (18-month-old) and older-aged (36-month-old) zebrafish were raised under a 14 h light/10 h dark cycle. Following dark adaptation to allow the separation of the retinal epithelium from the retina, their retinas were dissected as described below to isolate the retinal bipolar cells used to assess retinal structure and function, using whole-cell patch-clamping, IHC, and cellular imaging. All methodological procedures were performed in agreement with The University of Tennessee Health Science Center (UTHSC) Guidelines for Animals in Research. All procedures, including euthanasia (rapid chilling at 4 °C followed by decapitation) and tissue extraction methods, were reviewed and approved by the UTHSC Institutional Animal Care and Use Committee (IACUC; protocol #20-0170).

### 2.2. Retinal Bipolar Cell Isolation

Retinal bipolar cells were isolated from the harvested retinas using our established protocols [32,33,34]. Briefly, the retinas were harvested from the eyecups of dark-adapted zebrafish and treated with a saline solution containing 115 mM NaCl, 2.5 mM KCl, 0.5 mM CaCl_2_, 1 mM MgCl_2_, 10 mM HEPES, 10 mM glucose pH 7.4, and 1100 units/mL type V hyaluronidase (Worthington Biochemical Corp., Lakewood, NJ, USA) at 25 °C for 25 min. The retinal pieces were washed with saline, incubated in saline containing 5 mM DL-cysteine and 20~30 units/mL papain (Sigma-Aldrich, St. Louis, MO, USA), and gently triturated using a fire-polished glass Pasteur pipette. The resulting dissociated cells were plated onto glass-bottomed dishes in a saline solution containing 2.5 mM CaCl_2_. Mb1 ON-bipolar cells were identified on the basis of their characteristic morphology in the light microscope.

### 2.3. Bipolar Cell Voltage Clamp Recordings

Whole-cell patch-clamp recordings were made from bipolar cells that were acutely dissociated, as described previously [32,33,34]. The patch pipette solution contained 120 mM Cs-gluconate, 20 mM HEPES, 10 mM tetraethylammonium chloride, 3 mM MgCl_2_, 0.2 mM N-methyl-D-glucamine (NMDG)-EGTA, 2 mM Mg-ATP, and 0.5 mM GTP. The pipette solution was prepared with 35 ribbon-binding peptides (RBP) labeled with fluorescent 5-carboxytetramethylrhodamine (5-TAMRA) dye, custom ordered from LifeTein, LLC (Somerset, NJ, USA), and a low-affinity calcium indicator dye, Cal 520™ (AAT Bioquest, Inc., Pleasanton, CA, USA). Experiments designed to localize calcium signals to ribbon sites used a pipette solution that contained EGTA at 2 mM or 10 mM or the selective calcium chelator BAPTA at 2 mM. Calcium currents were evoked under a voltage clamp using an HEKA EPC-10 amplifier controlled by PatchMaster software version v2x90.4 (HEKA Instruments, Inc., Holliston, MA, USA). For all recordings, the holding potential was −65 mV and stepped to −10 mV for 10 ms. Membrane capacitance, series conductance, and membrane conductance were measured using the sine DC method of the PatchMaster lock-in extension and a 1600 Hz sinusoidal stimulus with a peak-to-peak amplitude of 10 mV centered on the holding potential [33]. The voltage clamp recordings were performed by placing the patch pipette in the terminal. The average series and membrane resistance for the voltage-clamp calcium current recordings were 36.5 ± 3.5 mΩ and 4.6 0.5 G Ω, respectively, and the average resting membrane capacitance was 2. 27 ± 0.04 pF for MA and 2. 73 ± 0.05 pF.

### 2.4. Laser-Scanning Confocal Microscopy

All fluorescence imaging was performed with a 60 × silicon objective (NA, 1.3) on an Olympus FV-3000 laser-scanning confocal microscopy system (Olympus, Shinjuku, Tokyo, Japan) or an Olympus IX-83 inverted microscope controlled by Olympus FV31S-SW software (Version 2.3.1.163) with a Galvano scanner. Acquisition parameters, such as pinhole diameter, laser power, PMT gain, scan speed, optical zoom, offset, and step size were kept constant between experiments. Sequential line scans were acquired at 3.02 ms/line and 10 μs/pixel with a scan size of 256 × 256 pixels. The time of image acquisition was confirmed via transistor–transistor logic (TTL) pulses between the FluoView software version 1.23.2.205 and the PatchMaster software version v2x90.4 used for patch-clamping (HEKA), in parallel with the voltage-clamp data.

### 2.5. Immunohistochemistry (IHC)

Eyes were harvested and sliced to visualize differences in the neuronal and ribbon structures of middle-aged and older-aged zebrafish retinas. The eyes were washed in phosphate-buffered saline (PBS), serially cryoprotected in PBS solutions containing increasing concentrations of sucrose (10%, 20%, and 30%), and embedded in optimal cutting temperature (OCT) medium (Fisher Healthcare, Houston, TX, USA). After being placed in a Tissue-Tek Cryomold (Fisher Scientific, Pittsburg, PA, USA), the eyes were snap-frozen and stored at −80 °C until use. They were subsequently cut with a Leica CM1850 cryostat (Leica Microsystems, Bannockburn, IL, USA) at −12–−15 °C into 15 μm sections, which were placed on Superfrost™ Plus microscope slides (Fisher Scientific) and allowed to dry at 25 °C for 2 h. A PAP pen liquid blocker (Electron Microscopy Sciences, Hatfield, PA, USA) was applied at the edges of the slides to create a hydrophobic barrier and air-dried for 15 min. The slides were washed thrice with PBS, permeabilized, blocked with Perm blocking solution (0.3% Triton-X-100 and 5% donkey serum in PBS), and incubated at 4 °C for 2 h in a hydrated chamber. The slides were incubated at 4 °C overnight with primary antibodies (Table 1) that had been diluted in a blocking solution. After washing with PBS, secondary antibodies conjugated to the indicated fluorophores (Table 2) were diluted 1:500 and applied to the slides, which were incubated at 25 °C for 2 h. The slides were washed once with PBS-Tween, thrice with PBS, and once with distilled water. Coverslips were attached using Prolong Diamond Antifade Mountant (ThermoFisher Scientific, Waltham, MA). IHC images were obtained with the aforementioned confocal microscope using the 60 × silicon objective with 1–7 × FluoView software (version 1.23.2.205) zoom and 0.45 mm z-step size. The acquisition parameters (laser power, PMT gain, scan speed, optical zoom, offset, step size, and pinhole diameter) were kept constant for each experimental dataset. Imaging and analysis were performed on the central retina across the group.

### 2.6. Data Analysis

Initial data analysis was performed in FluoView and PatchMaster, and the data were subsequently exported to Igor Pro (Version 8.04), Excel (Version 16.76), ImageJ (imagej.nih.gov accessed on 7 July 2023), and Imaris 9.9 (Version 9.9.0; Bitplane, Zurich, Switzerland) software for further analysis.

#### 2.6.1. Analysis of x-t Scans

To visualize the local calcium signals, the consistent acquisition parameters in ImageJ software (version 2.14.0/1.54f) were used to create a region of interest (ROI) that encompassed the ribbon (8 pixels, 331 nm), as described previously for tracking synaptic vesicle dynamics [32,34]. Individual Δ*F* images for the data sets described were normalized to the minimum value of Δ*F* for that image before generating averaged images.

#### 2.6.2. Analysis of Ribbon Synapses

The quantitative analysis of synaptic ribbons was performed in ImageJ and Imaris software. Images were deconvolved, and the manual cell counter plugins in ImageJ or automated spots module in Imaris software (Imaris 64, 9.9.0) were used to count the number of ribbons. Synaptic ribbon measurements were made using the surface rendering module of Imaris software to obtain multiple measurements from the same ribbon, and the longest length was used for between-group comparisons.

#### 2.6.3. Statistical Analyses

Statistical significance was assessed by using two-tailed, unpaired t tests with unequal variance using Microsoft Excel (Version 16.70) and Igor Pro 8 software (Wavemetrics, Lake Oswego, OR, USA). Variance in estimates of the population mean is reported as ± sem. Statistical significance of differences in average amplitudes of calcium current, capacitance, synaptic ribbon size and number, and calcium transients were assessed using unpaired, two-tailed t-tests with unequal variance.

## 3. Results

The bipolar ribbon synapses can transmit both tonic release and phasic release in response to sustained depolarization and play an important role in signaling changes in light intensity and overall luminance [21,22,23,24,25,26,27,28,29,30,31,35]. In this study, we primarily focused on comparing the brief release properties, local calcium signals, and synaptic ribbons. Notably, we limited our study to OA animals with no spinal curvature or cataract-like cloudy lenses.

Aging in humans, as well as in experimental animal models such as rats, mice, and zebrafish, is frequently associated with the gradual deterioration of sensory systems, including hearing and vision [4,5,6,13]. We hypothesized that aging might affect the phasic release properties in the terminals of bipolar cells and their ribbon synapses. Therefore, we measured the capacitance of OA and MA rod bipolar cell terminals to monitor exocytosis in response to evoked calcium currents with a pulse duration of 10 ms, which is sufficient to release the phasic component with little contribution from the tonic component (Figure 1). Shown are representative images of calcium currents and capacitance measurements for the MA (Figure 1A) and OA groups (Figure 1B). The average currents (Figure 1C) and membrane capacitance (Figure 1D) showed no changes between OA and MA groups, indicating that total release is maintained in response to 10 ms steps. Indeed, exocytosis efficiency shows no changes between OA and MA, supporting our findings that total release is maintained in response to 10 ms steps (Appendix A).

The synaptic ribbons are specialized organelles found in multiple sensory cells, including the inner hair cells (IHCs) that are involved in the detection and amplification of sound waves in the auditory system. Strikingly, aging results in a drastic loss of IHC synaptic ribbons [5,36]. In the visual system, synaptic ribbons comprised primarily of the ribeye protein play a crucial role in maintaining proper vision [37,38,39]. To examine whether aging had any effect on the number or morphology of ribbon synapses in bipolar cell terminals, we visualized the ribbons by patch clamp physiology and confocal microscopy. To do so, we performed the whole-cell patch clamping of bipolar cell terminals of MA and OA zebrafish (Figure 2A,B, respectively) using a patch pipette containing 35 mM TAMRA-labeled ribeye binding peptide (TAMRA-RBP) as described [40] and localized the synaptic ribbons by simultaneous fluorescence confocal microscopy, as we described previously [33,34]. We observed a rapid increase in RBP fluorescence and bright fluorescent spots that indicated the location of the ribbon.

The ribbons detected in these experiments were quantified in a series of z-axis optical sections through the entire terminal using ImageJ and Imaris software, as described in Materials and Methods. The surface module of Imaris software was used to generate representative three-dimensional (3D)-reconstructed models of bipolar cell terminals from MA or OA zebrafish, in which the ribbons were labeled with TAMRA-RBP (Figure 2C,D). The mean total number of ribbons (mean values ± SEM) in a bipolar cell synaptic terminal from OA fish (15.1 ± 0.9; *n* = 19 cells in seven animals; Figure 2E) was significantly lower (*p* < 0.001) than those in the synaptic terminals from MA fish (37.7 ± 0.7; *n* = 19 cells in seven animals; Figure 2E).

Interestingly, we also observed a significant increase (*p* < 0.001) in the length of the synaptic ribbon (0.57 ± 0.01 mm; *n* = 227 ribbons) in the bipolar cell terminals of OA zebrafish, relative to those in the bipolar cell terminals of MA fish (0.42 mm ± 0.01; *n* = 300 ribbons, as shown in Figure 2F). These findings suggest that the bipolar cell terminals of OA fish may compensate for the decreased number of ribbons by increasing the lengths of the individual ribbons, or they could be an aging factor.

To determine whether the changes in the number and size of synaptic ribbons observed in single Mb1 bipolar cells from OA zebrafish alter the morphology of the retinal inner plexiform layer (IPL), we double immunostained retinal cryosections from MA and OA zebrafish with fluorescently labeled antibodies specific for ribeye a and for the rod bipolar cell marker protein kinase C (PKCα), as described [41,42,43]. The patterns of PKCα staining in the retinas of MA and OA zebrafish are shown in Figure 3A,B, respectively. We found that a mouse monoclonal anti-PKCα antibody (A-3; sc-17769) labeled a subset of ON-bipolar cells with large axon terminals (Figure 3A, top panel, white arrow) and a separate subset with smaller axon terminals (Figure 3A, top panel, white arrowhead) that ramify the more proximal and distal parts of the IPL, respectively. Based on previous studies, these two subtypes likely comprise (1) the B_ON_ s6L or RRod type of ON-bipolar cells that morphologically resemble the mixed-input (b1 or Mb; (hereafter we call Mb1)) ON-bipolar cells of other teleost fish [41] and (2) the B_ON_ s6 type that contact the cones (named cone-ON) [41,44,45]. Unlike the hexagonal shape of the bipolar cell soma we found in 5 dpf zebrafish larvae [43], we observed a subset of PKCα-labeled bipolar cells from MA and OA zebrafish that exhibited the pear-shaped morphology described for adult Mb1 rod bipolar cells [46]. As expected, these appear to connect to the larger terminals of the Mb1-type bipolar cells (Figure 3A, top panel, labeled with white # symbols). Thus, these cells appear to be associated with both the scotopic (dark-adapted) and photopic (light-adapted) pathways that involve rods and cones, respectively.

The quantitative analysis of the PKCα-expressing subset of ON-bipolar cells revealed distinct changes in the Mb1 vs. cone-ON bipolar cells and OPL-IPL laminar structure as described below. First, there appeared to be fewer Mb1 bipolar cells in the retina of OA zebrafish than in those from MA fish (MA 1.32 ± 0.1 vs. OA 0.48 ± 0.1 in a 370 mm^2^ region of interest; *n* = 2~4 sections in four retinas from two fish; *p* < 0.001). However, we observed no changes in the number of cone-ON bipolar cells in the OA versus the MA retinas (MA 7.18 ± 0.4 vs. OA 6. 78 ± 0.3 in a 370 mm^2^ ROI; *n* = 2~4 sections in four retinas from two fish). Next, the soma size of the Mb1 bipolar cells from OA zebrafish was significantly larger (*p* < 0.006) than the average length of the long axis of Mb1 bipolar cells from MA fish (MA 9 ± 0.2 mm vs. OA, 12.3 ± 0.3 mm; *n* = 2 four retinas from two fish), while we observed no differences in the size of cone-ON bipolar cells isolated from MA or OA zebrafish (MA, 6.2 ± 0.2 mm vs. OA, 6.0 ± 0.2 mm; *n* = 2~4 sections in four retinas from two fish).

Interestingly, the IPL that encompasses the axon length and terminal area of the Mb1 and cone-ON bipolar cells was significantly thinner (*p* < 0.001) in OA zebrafish than that in the MA fish (Mb1: MA, 60.5 ± 1.62 mm, OA, 44.5 ± 0.9 mm vs. Cone-ON: MA, 44.9 ± 1.3 mm, OA, 32.35 ± 1.5 mm, *n* = 2~4 sections in four retinas from two fish), as shown in Figure 3A,B. The quantification of retinas stained with ribeye a-specific antibodies revealed that the IPL ribbon clusters were significantly shorter (*p* < 0.007) in the OA zebrafish than that in the MA fish (MA, 37.2 ± 2.4 mm vs. OA, 25.5 ± 0.7 mm; *n* = 2~4 sections in four retinas from two fish), as shown in Figure 3C,D. In contrast, we observed no differences in the colocalization area of the ON bipolar cells stained for PKCα and ribeye a in OA and MA fish (MA, 10.3 ± 1.9 mm vs. OA, 10.9 ± 1.1 mm; *n* = 2~4 sections in four retinas from two fish). These findings suggest that the substantial changes observed in the INL and IPL of OA zebrafish can be correlated with the Mb1 rod bipolar cells that govern scotopic information.

Ca^2+^ influx in bipolar cells has been shown to occur preferentially in the ribbons [33,47,48]. Thus, to explore any functional implications associated with the observed changes in the number and size of synaptic ribbons observed in single Mb1 bipolar cells, we conducted patch-clamping experiments and compared local calcium signaling of a single ribbon as we described previously (Materials and Methods) [33]. Briefly, bipolar-cell terminals were filled via a whole-cell patch pipette with TAMRA—RBP to mark ribbons and the low-affinity calcium indicator, Cal520LA, (AAT Bioquest Inc., Pleasanton, CA, USA) to monitor changes in [Ca^2+^]_i_ using rapid x-t line scans at ribbon locations from MA and OA Mb1 bipolar cells (Figure 4A,B). The excitation laser was scanned along the line defined perpendicular to the plasma membrane at a ribbon, extending from the extracellular space to the cytoplasmic region beyond the ribbon at a rate of 3.1 ms per line.

Figure 4A,B shows individual examples of Cal520LA fluorescence intensity vs. time (horizontal axis) and distance (vertical axis) at a ribbon location, together with RBP fluorescence, to indicate the position of the ribbon along the scanned line obtained from MA and OA Mb1 bipolar cell synaptic terminal. Data points in Figure 4C,D are the spatially averaged intensity in each scan line across three 10 ms depolarizations (onset at arrow) at ribbon locations. In MA Mb1, during the stimulus, fluorescence rose more rapidly and to a higher level at the ribbon than that of OA, indicating that the number of calcium channels clustered at the ribbon in OA is likely to be altered. After depolarization terminated, Cal520LA fluorescence at the ribbon quickly collapsed to the same basal level of Ca^2+^. These results provide evidence for changes in the local domain of high Ca^2+^ driving rapid vesicle fusion, specifically at the ribbon location in OA Mb1 rod bipolar cells.

Studies in retinal bipolar neurons have used exogenous calcium chelators such as ethylene glycol-bis(β-aminoethyl ether)-N,N,N′,N′-tetraacetic acid (EGTA) and 1,2-bis(o-aminophenoxy)ethane-*N*,*N*,*N*′,*N*′-tetraacetic acid (BAPTA) to localize the Ca^2+^ signals to near the Ca^2+^ entry [30,47,48,49,50,51,52]. Thus, to investigate the ribbon-associated calcium signals in the Mb1 bipolar cells from MA and OA zebrafish, we increased the EGTA concentration in the pipette solution to 2 mM or 10 mM or included 2 mM BAPTA and compared the local calcium signals, as described for Figure 4. EGTA binds calcium slower but has a higher affinity for Ca^2+^ than Cal520LA and, thus, is expected to compete poorly with the fast-binding Ca^2+^ indicator initially as Ca enters the cell but is expected to outcompete the indicator at equilibrium. Because of this, Cal520LA signals in the presence of excess EGTA preferentially reflect signals near Ca^2+^ channels [33,47]. We found that, at both concentrations of EGTA, OA fish showed lower calcium transients than MA fish (Figure 5A–D). BAPTA, which is both a fast and high-affinity buffer, nearly eliminated the local calcium transient in the MA bipolar cell terminal (Figure 5E) and OA bipolar cell terminal (Figure 5F). These findings suggest that aging is accompanied by changes in the local calcium transients at individual bipolar synaptic ribbons, likely due to the mislocalization of calcium channels.

## 4. Discussion

The visual decline often associated with aging impacts the morphology and overall function of the retina, inner hair cells, horizontal cells, retinal ganglion cells, Müller glial cells, and photoreceptors [5,53,54,55]. Some age-related morphological changes have also been observed in rod bipolar cells using different animal models, as described below. The zebrafish model is appropriate for aging studies because these animals share a 70% genomic similarity with humans, and their short life span provides the unique ability to study the progression of aging since they often display aging markers that are similar to those observed in humans, including spinal curvature, cognitive decline, and visual impairment. To the best of our knowledge, this is the first study to examine the changes in the retinal bipolar cells of older-aged (OA) zebrafish.

We found that the number of synaptic ribbons in bipolar cells isolated from OA zebrafish retina was reduced but that the ribbons that were present exhibited an increased length, suggesting a compensatory mechanism for the decreased number of ribbons in the bipolar cell terminals of OA, or they could be consequences of aging. We also detected a significant reduction in the synaptic ribbon expressed in the IPL in the OA zebrafish retina, as detected by staining with antibodies specific for the ribbon protein ribeye a. Antibodies specific for protein kinases α (PKCα) and β (PKCβ) have been used to identify retinal ON-bipolar cells in different species [56,57,58,59,60,61,62,63,64]. PKCα is broadly used as a marker for rod bipolar cells in mammals and the corresponding mixed-type ON-bipolar cell (Mb1) in teleost fish, including zebrafish [41,42,56,57,58,59,60,61,62,63,64]. The PKCα antibodies we used for our IHC assay detected two populations of bipolar cells differentiated by the size of their axon terminals (large and small) in the retinas of both MA and OA zebrafish that are likely to correspond to the B_ON_ s6L ON-bipolar cell that is identical to the Mb1/RRod cell that contacts only rods and the B_ON_ s6 ON-bipolar cell type that contacts cones, as described previously [41,44,45]. The quantitative analysis of the IHC assay data showed a significant loss of Mb1 but not of cone-ON bipolar cells. However, we found that the sizes of the soma and terminals of Mb1, but not cone-ON bipolar cells, were larger in bipolar cells from OA fish than those from MA fish, while the length of both Mb1 and cone-ON bipolar cells was significantly reduced in the OA zebrafish retina. These data show substantial alterations on PKCα labeled ON bipolar cells and IPL ribbon structures that are associated with the older fish population. Of note, the resting capacitance for OA zebrafish is higher than MA is consistent with our findings that the size of soma and terminals are larger in OA. We believe that it reflects the fact that fish retinas grow throughout the life of the animal, and the cells probably just get larger as part of this.

In humans, aging is related to an overall reduction in the thickness of individual retinal layers, except for the foveal retinal nerve fiber layer (RNFL) and the inner and outer segments of the photoreceptors, which significantly increase with age [65]. The thickness of the mouse retina decreases by 15% with age but is compensated for by a corresponding 15% increase in the retinal area [6]. In the same study, the size and complexity of overall arborization decreased with age, as did connections in the IPL across different synaptic types [6]. Importantly, rod bipolar cell dendrites behaved differently with age, expanding out of the OPL and into the ONL [6]. A different study of the mouse retina observed that, while the rod bipolar cells of younger mice are restricted to the OPL, those of old animals extend into the ONL [66]. On the contrary, in diurnal Chilean Degu (*Octodon degus*), significant age-related degeneration was observed in rod bipolar cell dendrites, which became retracted, more flattened, and less branched with advanced age [10]. The rod bipolar cells of older Chilean rodents are also reduced in density, with significantly smaller terminals than in those of younger rodents [10].

As we described, our observations matched the previous mouse findings of retinal layer thinning, with the IPL becoming significantly reduced in the aged zebrafish. Similar to in the diurnal Chilean rodents, aged zebrafish rod bipolar cells were reduced in number and exhibited significantly lesser branching than those from younger fish. However, unlike the Chilean rodents, zebrafish showed an increase in the size of both the terminals and the soma of rod bipolar cells. Our observations may present an indication that, while aging contributes to the progressive degradation of rod bipolar cells and the integrity of retinal layering, the individual cells manifest a significant degree of plasticity, which we interpret as an attempt to compensate for the reduction in the overall size of the structure. We also observed a similar pattern in the number and size of synaptic ribbons within the terminals of rod bipolar cells. Aged zebrafish display fewer synaptic ribbons in the terminals of their rod bipolar cells, which possess greater lengths than those of their middle-aged counterparts. Synaptic ribbons are essential for synaptic transmission in sensory systems [37,38,39,67,68] and are attached to the presynaptic plasma membrane close to voltage-gated calcium channels, and they tether multiple synaptic vesicles [52,69,70,71,72,73,74,75,76,77,78]. In the aging human retina, swollen and floating synaptic ribbons are accompanied by abnormal sphere-shaped ribbon synapses in parafoveal cone terminals [54]. Studies on the mouse cochlea report that the number of ribbons in inner hair cells was significantly reduced with aging but that the loss of the synaptic structures was accompanied by a compensatory increase in the volume of the remaining ribbons [5]. Similar to the observations made in other cell types, we observed that the age-related loss of ribbon structures was accompanied by an increase in length. As previously described, this balance suggests a possible compensatory mechanism that might account for the conservation of normal retinal function despite these age-related alterations.

Our live imaging of local calcium signals at a single synaptic ribbon showed that the amplitude of calcium signals was significantly lower in OA than in MA zebrafish. Despite these local changes at a single ribbon, we found no overall changes in the evoked calcium current and brief release properties measured as a function of capacitance. These findings suggest the possible existence of as-yet unknown compensatory mechanisms that might help compensate for the loss of ribbons in OA zebrafish. Of note, individual Mb1 bipolar cells signal to multiple post-synaptic cells via synapses with different kinetics [28,30], and circuit function may be altered despite the lack of change in total fast release. In mice, aging results in the enlargement of presynaptic auditory ribbons, similar to those we reported here, and such change is accompanied by increases in presynaptic calcium signaling related to a stronger sustained exocytotic response [36]. In other studies, although the inner hair cells of aging mice contain fewer but larger ribbon synapses, the size and kinetics of exocytosis and vesicle replenishment are unaffected by age [5]. The lack of changes in calcium currents in aged fish in response to brief pulses suggests that the number of calcium channels in the terminals of the rod bipolar cells may remain relatively constant between the age groups we examined. Likewise, the unchanged capacitance measurements and exocytosis efficiency suggest that the number of vesicles that are fused during exocytosis is also likely to remain consistent. We found that the exocytosis efficiency in MA zebrafish for brief stimuli (0.98 ± 0.14) is lower compared to previous findings that reported hair ribbon synapses in larval zebrafish in response to longer/sustained stimuli, approximately ~1.42. These differences could be due to differences in the age of the fish, pipette solution, calcium channel subunits, number of vesicles associated with synaptic ribbons, and strength of stimuli. In light of the crucial role played by ribbons in synaptic transmission and our observed decrease in the number of ribbon structures in the terminals of rod bipolar cells and the local calcium transient, we expected to detect functional changes in the retinas of the older-aged zebrafish. However, it is plausible that the lengthening of the existing structures compensated for the reduced number of ribbons and led to an increase in the number of synaptic vesicles tethered near the membrane on the remaining ribbon(s), which could explain the lack of changes in exocytosis we observed. Another possibility is that, in the OA fish, synaptic vesicles may exhibit a higher affinity for calcium than those in the MA fish, resulting in the pattern of responses to stimulation we detected in the MA fish. This could indicate the existence of an adaptive mechanism in older fish, in which they might, for example, require a lower amount of calcium to maintain regular functioning than would a younger zebrafish.

Although we found that the global calcium currents in aged zebrafish remained unchanged, we detected a significant decrease in the local calcium transients that we hypothesized is associated with aging. Calcium dynamics are essential for vesicle release and regulate distinct steps in the process. Thus, we measured calcium levels via a calcium transient that reflects localized calcium levels near the proximal region of the ribbon and analyzed the effect on calcium dynamics by adding EGTA or BAPTA to the cell during patch clamping. These buffers are calcium chelators that compete with endogenous calcium sensors, thereby reducing the free calcium concentration of the sensor [28,30,79]. EGTA and BAPTA have similar binding affinities for calcium but different kinetics, with BAPTA binding calcium approximately 40 times faster than EGTA [79]. While previous studies defined the overall structural changes and functional responses of ribbon synapses in aging [5,10,36,53,80,81], to the best of our knowledge, our study is the first to report the effect of aging on local calcium dynamics. In the presence of EGTA, we observed a significant reduction in local calcium transients in aged fish relative to middle-aged fish. These findings suggest that the local calcium concentration is significantly lower in aged fish. Given that the calcium current remains unaltered between the age groups we studied, this observation supports the possibility that calcium channels might not be clustered at the ribbon’s proximal region but instead could be distributed extrasynaptic. This more even distribution of calcium channels could contribute to the continued function of the ribbon synapses, even in the presence of locally altered calcium dynamics.

Given that we detected local changes in the rod bipolar cells of aging zebrafish, the observed alterations in vision may be attributed to other factors that might differ between middle-aged and older-aged retinas, for example, proteins or non-ribbon release mechanisms. Extensive studies from various species, including goldfish, mice, and salamanders, demonstrated that, while brief stimuli primarily trigger events close to the ribbons, supplemental vesicle release also occurs at more distant regions, where both ribbon and non-ribbon release sites are used [22,47,68,82,83,84]. Within the cells we tested, we hypothesized that in older fish, the cells we examined may possess a greater ability to receive support from non-ribbon release sites during brief stimulation than middle-aged fish. Although sustained depolarization intensifies non-ribbon release due to calcium spreading throughout the terminals, even brief stimulation could contribute to an additional non-ribbon release [84], which lends support to our hypothesis. Another potential factor influencing the observed local changes is the presence of different synaptotagmin (Syt) isoforms in bipolar cells [85,86]. For example, Syt7 is selectively crucial for asynchronous and delayed release triggered by calcium but not for calcium-dependent replenishment in retinal rod bipolar cell synapses [85]. In aged fish rod bipolar cells, alterations in the expression levels or activity of Syt7 could lead to changes in the timing or magnitude of asynchronous release, which would help to explain the local changes in synaptic vesicle release that we observed.

Because zebrafish live approximately three years in the laboratory setting, the subsequent limited availability of aged fish for experimentation constrained the types of experiments and numbers of trials that could be performed. For example, we focused on experiments with brief individual pulses (10 ms), which would address only the release of the proximal vesicle pools within the ribbon. Additionally, while the regenerative abilities and constant growth of zebrafish [15] might render such studies less applicable to humans, these regenerative abilities decrease with age [87]. If the observed changes were significant in an animal with regenerative capability, the effects might be even more pronounced with aging in a species that lacks such capacity, perhaps resulting in clearer signs of visual impairment. Further, in zebrafish and goldfish, ribbons from bipolar cells and photoreceptors are known to disappear at night and reappear in the morning. This diurnal synaptic plasticity of ribbons with circadian rhythm may become impaired with age [88]. It will be interesting in the future to examine whether this putative impairment may thus cause the effects of aging.

The reduction of IPL observed in our studies has also been reported during pathology. For example, in Parkinson’s disease, the GCL-IPL complex of the macula demonstrated thinning even in the earliest stages [89]. Of note, the loss of synaptic ribbons and changes in the synaptic ribbons are likely to alter synaptic counterparts, as discussed previously in hair cell ribbon synapses. For example, in goldfish, repetitive activation of Mb1 cells was shown to rapidly augment the synaptic strength of a subgroup of amacrine cell synapses [90]. Further, in zebrafish hair cells, loss of synaptic ribbon terminal responsiveness following AMPA exposure was demonstrated due to the loss of postsynaptic dysfunction [91]. Finally, calcium dysregulation is a common factor altered in any age-related disease, particularly in Alzheimer’s brains and retinas [92,93,94]. Thus, the altered calcium we see in normal aging can likely alter vision.

To gain a more comprehensive understanding of the effects of aging on vision and to define their mechanisms, future studies will explore the effects of different pulse durations and more complex pulses on calcium dynamics. Finally, future studies that explore the roles of synaptotagmins, other proteins, and non-ribbon mechanisms related to aging processes will provide deeper insights into the complex calcium dynamics involved in aging and vision.

## 5. Conclusions

We showed that zebrafish of advanced age acquired changes in their synaptic ribbon structure and local calcium dynamics, thereby providing valuable insight into the morphological and functional alterations in the aging retina, specifically in rod bipolar cells and their ribbon synapses. These findings suggest that, while normal aging may not significantly impact the deterioration of vision, there may be more complex visual changes occurring that contribute to the visual impairment observed in human adults. The subtle changes we observed may have significant implications for disease models in which such alterations may be amplified, potentially resulting in visual impairments. The present study contributes to the growing knowledge of aging-associated changes in the visual system and will facilitate further studies to explore the implications of calcium regulatory mechanisms for age-related visual disorders.

## Figures and Tables

**Figure 1 cells-12-02385-f001:**
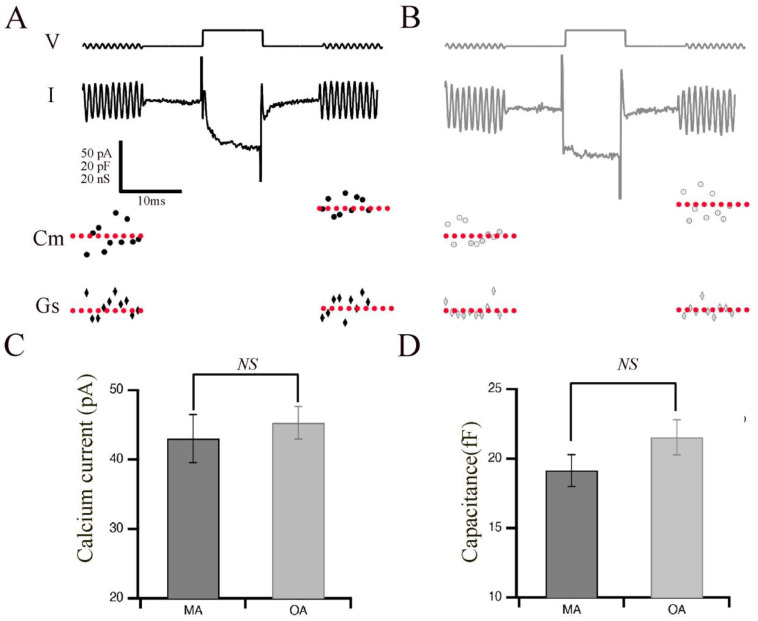
Retinal bipolar cells isolated from old-aged zebrafish exhibited no changes in their brief release properties relative to middle-aged fish. (**A**,**B**) Ca^2+^ current (I) recorded from the synaptic terminal of a bipolar neuron isolated from middle-aged (MA; panel (**A**)) and older-aged (OA; panel (**B**)) zebrafish in response to a voltage-clamp pulse (V) from −60 mV to −15 mV for 10 ms. The sinusoidal voltage stimulus used to monitor membrane capacitance (C_m_) and series conductance (G_s_) is visible at the beginning and end of the voltage trace. Below the traces are the resulting values of C_m_ for MA (filled circles) vs. OA (open circles) fish and of G_s_ for MA (filled diamonds) vs. OA (open diamond) fish before and after the activation of the Ca^2+^ current. (**C**,**D**) Average calcium current (**C**) and capacitance (**D**) in response to a voltage-clamp pulse (V) from −60 mV to −15 mV for 10 ms that was obtained from bipolar neurons of MA (**A**) and OA (**B**) zebrafish. *n* = 20 MA, seven animals: *n* = 12 OA bpcs; nine animals. Individual values of Ca^2+^ current capacitance measurements are shown in Appendix A.

**Figure 2 cells-12-02385-f002:**
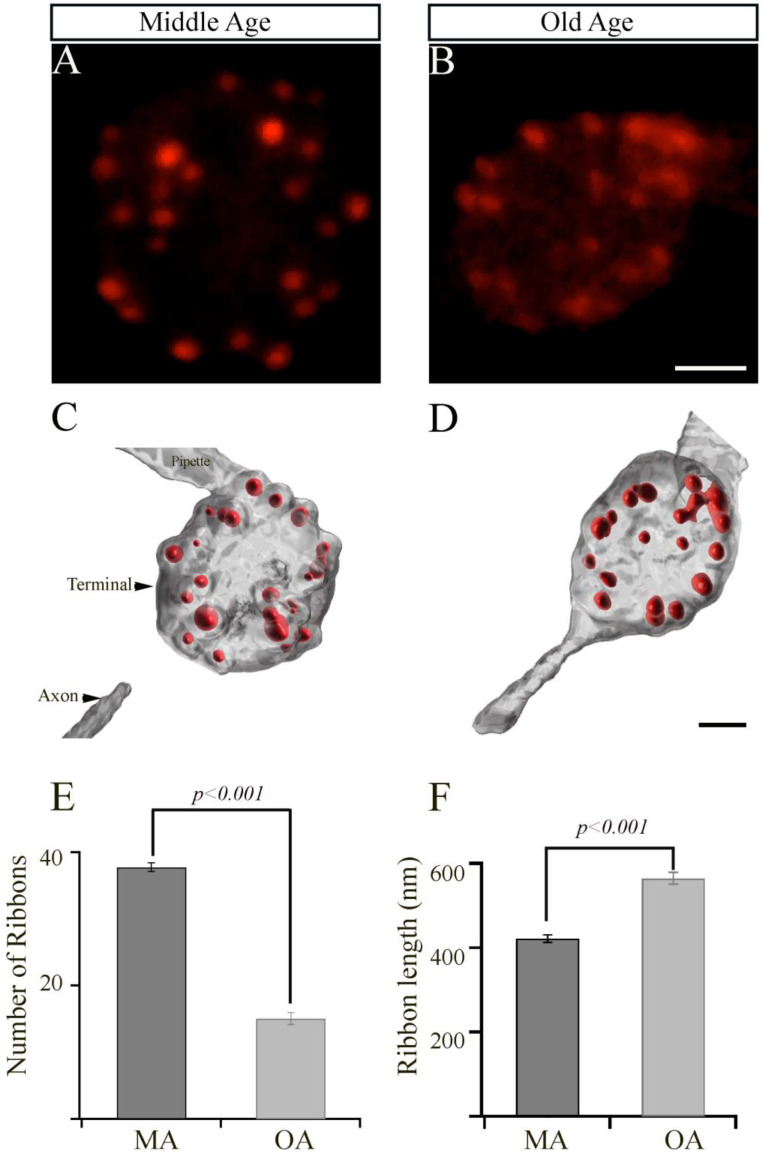
The retinal bipolar cells isolated from older-aged zebrafish exhibit changes in the numbers and length of their synaptic ribbons. (**A**,**B**) Representative two-dimensional projections of the bipolar cell synaptic terminals from middle-aged (MA; panel (**A**); *n* = 19 cells in seven animals) and older-aged (OA; panel (**B**); *n* = 19 cells in seven animals) zebrafish in which the synaptic ribbons were labeled by voltage clamping with a whole-cell pipette, whose internal solution contained fluorescent TAMRA-RBP peptides and was visualized by confocal microscopy. Scale bar, 2 µm. (**C**,**D**) Imaris-generated three-dimensional (3D)-reconstruction of zebrafish bipolar cells whose synaptic ribbons were labeled with TAMRA-RBP (red) and visualized as described above. Scale bar, 5 µm. (**E**,**F**) The average number of ribbons (**C**) and the ribbon lengths (**D**) contained by zebrafish bipolar neurons isolated from MA (**A**) or older-aged (**B**) zebrafish as described in Section 2.

**Figure 3 cells-12-02385-f003:**
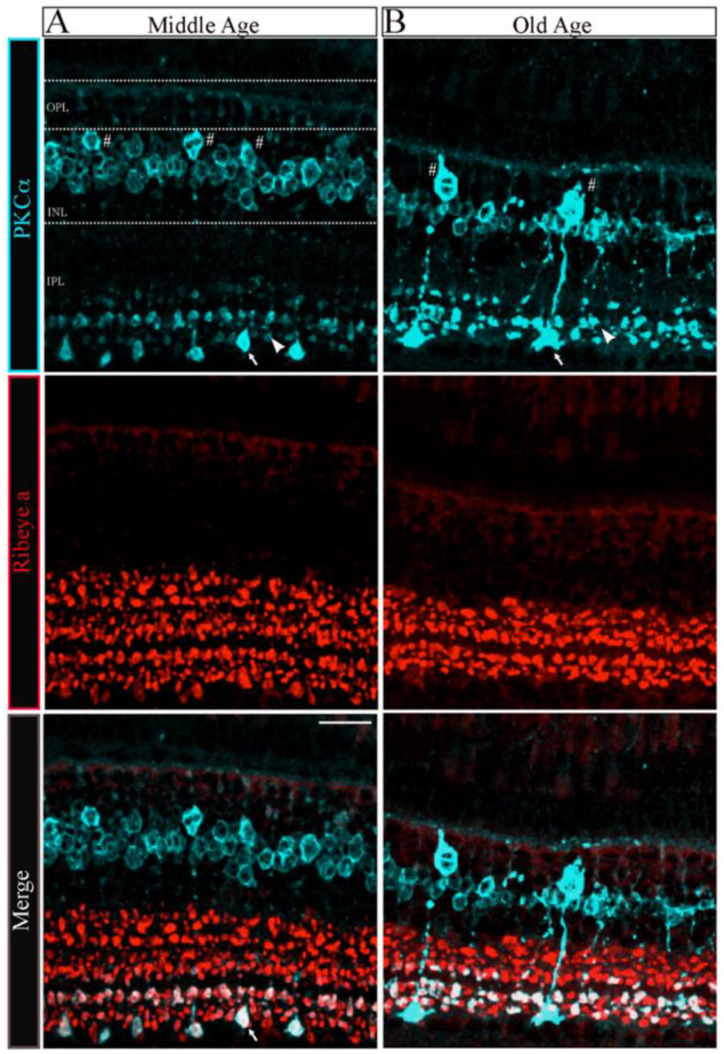
The morphology of bipolar cell ribbon synapses in the retinal IPL was altered in older-aged zebrafish. (**A**,**B**) Transverse retinal sections from middle-aged ((**A**), left panels; *n* = 2~4 sections in four retinas from two fish) and old-aged ((**B**), right panels; *n* = 2~4 sections in four retinas from two fish) zebrafish were double immunostained with fluorescently labeled antibodies specific for the rod bipolar cell marker PKCα (cyan; **top panels**) or ribeye a (red; **middle panels**); also shown is the overlay of PKCα and ribeye a labeling (merge; **bottom**). Maximal intensity projections are shown, and the relative positions of the INL and IPL are indicated. Scale bar, 20 µm. PKCα, protein kinase C alpha; OPL, outer plexiform layer; INL, inner nuclear layer; IPL, inner plexiform layer. Larger (arrow) and smaller (arrowhead) are indicated in PKCα ribbon terminal morphologies in MA and OA. # Denotes the pear-shaped soma characteristics of Mb1 rod bipolar cells with larger terminals.

**Figure 4 cells-12-02385-f004:**
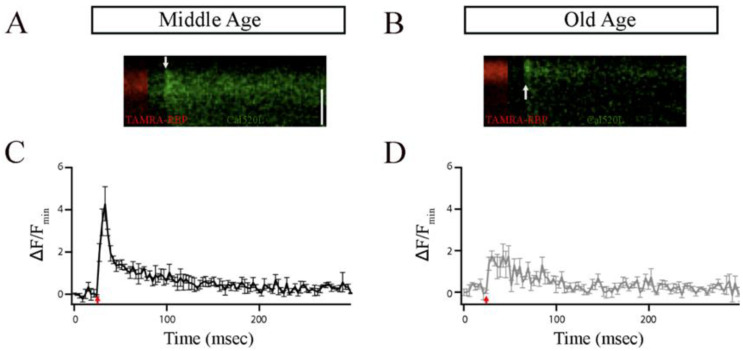
The ribbon synapses in the Mb1 bipolar cells from older-aged zebrafish exhibited altered Ca^2+^ responses after brief depolarization. (**A**,**B**) The *x*-*t* plots show the Cal520LA fluorescence intensity (green staining in the right section of each plot) at a single ribbon location as a function of time (horizontal axis); RBP-TAMRA fluorescence (red staining in the left section of each plot) indicates the position of the ribbon along the scanned line, while the darker region at the top of each plot is the extracellular space. White arrows indicate the timing of depolarization. (**C**,**D**) Spatially averaged Cal520LA fluorescence as a function of time at Mb1 bipolar cell ribbon from middle-aged (MA; panel (**C**)) or older-aged (OA; panel (**D**)) zebrafish. Shown is the average intensity (±SEM) in each horizontal row of pixels for three separate 10 ms depolarizations with similar calcium currents (41 ± 4 pA; *n* = 3 cells; three fish). Fluorescence intensity was normalized by the baseline fluorescence before stimulation by averaging over all pixels (i.e., over space and time) and dividing by the baseline fluorescence (*F*_min_). The red arrow indicates the onset of the 10 ms depolarizing stimulus.

**Figure 5 cells-12-02385-f005:**
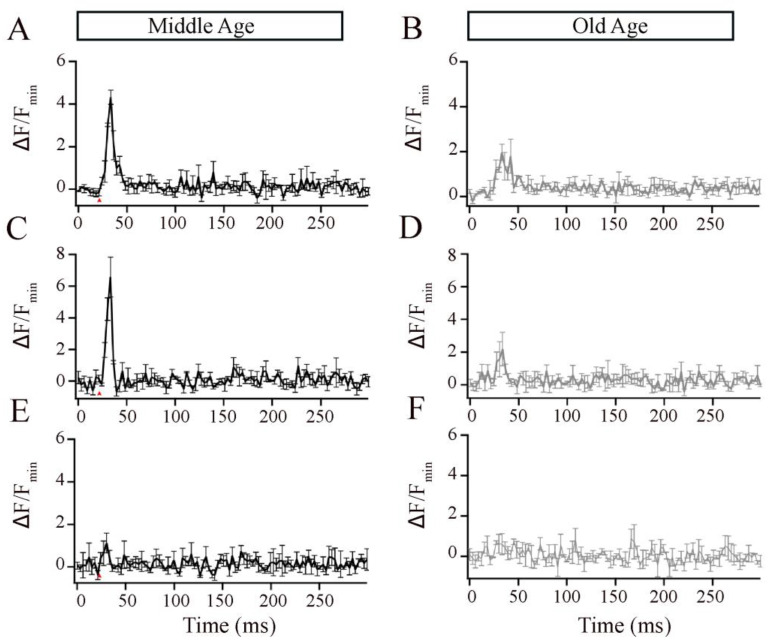
The local Ca^2+^ responses elicited by brief depolarization at an Mb1 bipolar cell synaptic ribbon were altered in older-aged zebrafish. (**A**,**B**) Average Cal520-2LA fluorescence at ribbon locations in response to 10 ms depolarization in Mb1 bipolar cells isolated from middle-aged (MA; left panels; *n* = 3 ribbons from 3 cells and fish) and older-aged (OA; right panels; *n* = 3 ribbons from 3 cells and fish) zebrafish with pipette solution containing 2 mM EGTA. Shown is the change in fluorescence (Δ*F*) from baseline before stimulation, divided by the baseline fluorescence (*F*_min_). (**C**,**D**) Same as in (**A**,**B**), except that the pipette solution contained 10 mM EGTA. (**E**,**F**) Same as in A-B, except that the pipette solution contained 2 mM BAPTA. Red arrows indicate the time of pulse stimulation.

**Table 1 cells-12-02385-t001:** Primary antibodies used for IHC.

Antigen	Antiserum	Host	Dilution	Source (Number)	Marker for
PKC	Monoclonal anti-PKC	Mouse	1:250	Santa Cruz (sc-17769)	Rod-type bipolar cells
Ribeye-A	Polyclonal anti-ribeye	Rabbit	1:1000	Zenisek lab (s4561-2)	Synaptic ribbons (IPL)

**Table 2 cells-12-02385-t002:** Secondary antibodies used for IHC.

Antibody	Conjugation	Source (Number)
Donkey anti-guinea pig	Cy3	Jackson Immunoresearch, #706-165-148
Donkey anti-mouse	Alexa Fluor 647	Southern Biotech, #6440-31

## Data Availability

Available on request.

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
