# Peer review of "The Effects of Aging on Rod Bipolar Cell Ribbon Synapses"

_cells, 2023, doi:10.3390/cells12192385_

Round 1

Reviewer 1 Report

The manuscript on title “The effects of Aging on Rod bipolar cell Ribbon synapse” is well structured and is systematically written. Scientific experiments are convincing with conclusion authors have made based on the results, but still there are few points that I will like authors to answer either actual experiment or with scientific explanation.  

  1. Authors have mentioned in the Fig 2 that they have observed increase length of SR (synaptic ribbon) in OA model, do they think this is compensatory mechanism for the loss of ribbon/s. Also, how will they explain this phenomenon with Knockout models of SR.  
  2. Is there a loss of bipolar cells show in the Fig 3, is it possible to document it? 
  3. Is it possible to show loss of SR in Fig 2 and how does the distribution of Synaptic vesicle look at Electron microscopy level? 
  4. Was there a distinct change in the rod and cones current and exocytosis and is it possible to show by plot by showing rod and cones separately (Comment for the Fig1)?  
  5. What is the explanation of increase soma size for Mb1 Bipolar cells? 
  6. Will author like to comment on basal level of calcium in OA model, do they see any changes in it.
  7. Although subtle changes (Fig1) were observed in the calcium current of bipolar cells still it is important to show if the calcium channels are miss-localized or how they look in terms of their localization with SR  
  8. Does calcium seem to be for critical of aging defects in the bipolar cells if not what other factors may be contributing towards physiological changes in the aging retina? what are authors insight on it.  

Author Response

Reviewer 1

The manuscript on title “The effects of Aging on Rod bipolar cell Ribbon synapse” is well structured and is systematically written. Scientific experiments are convincing with conclusion authors have made based on the results, but still there are few points that I will like authors to answer either actual experiment or with scientific explanation.  

Response: We appreciate the time and energy the reviewer committed and the value of their comments. We have incorporated our comments to the best of our knowledge.

  1. Authors have mentioned in the Fig 2 that they have observed increase length of SR (synaptic ribbon) in OA model, do they think this is compensatory mechanism for the loss of ribbon/s. Also, how will they explain this phenomenon with Knockout models of SR.  

Response: Thank you for asking the important question. We believe this could be a compensatory mechanism or an aging factor. We have mentioned this point in the results (please see lines 262-263) and now have added it to the discussion (please see lines 405-408). It is difficult to compare to the Ribeye knockout animals, since those animals lack ribbons altogether and are unable to either change the size or number of ribbons.

  1. Is there a loss of bipolar cells show in the Fig 3, is it possible to document it? 

Response: Thank you for the great question. We documented changes in the number of bipolar cells in the results (lines 297-301): “fewer Mb1 bipolar cells in the retina of OA zebrafish than in those from MA fish (MA 1.32±0.1 vs. OA  0.48±0.1 in a 370 mm2 region of interest; n=2~4 sections in four retinas from two fish; p<0.001). However, we observed no changes in the number of cone-ON bipolar cells in the OA versus the MA retinas (MA 7.18±0.4 vs. OA 6. 78±0.3 in a 370 mm2)”. As per these findings, we believe the reduction could account for possible loss. We also described this information in the discussion as “Quantitative analysis of the IHC assay data showed a significant loss of Mb1 but not of cone-ON bipolar cells” (lines 419~423).

  1. Is it possible to show loss of SR in Fig 2 and how does the distribution of Synaptic vesicle look at Electron microscopy level? 

Response: This is a very important question we plan to address. As mentioned in our 1st submission, we had a limited number of fish for experiments and are unable to do additional electron microscopy experiments at this time. Thanks again for the great advice.

  1. Was there a distinct change in the rod and cones current and exocytosis and is it possible to show by plot by showing rod and cones separately (Comment for the Fig1)?  

Response: Our focus in this paper is on ON-type bipolar ribbon synapses, which include Mb1 rod bipolar cells and ON-con bipolar cells. All electrophysiology and calcium imaging measurements are focused on Mb1 rod bipolar cells. Rods and cones are beyond the focus of this paper. Recordings from rods and cones would be a significant undertaking that we feel is beyond the scope of this paper.

  1. What is the explanation of increase soma size for Mb1 Bipolar cells? 

Response: Thank you for the great question. We believe that it reflects the fact that fish retinas grow throughout the life of the animal, and the cells probably just get larger as part of this. We have added this explanation to the discussion now (lines 426-428).

  1. Will author like to comment on basal level of calcium in OA model, do they see any changes in it.

Response: Since the calcium measurements are not ratiometric, we are cautious about commenting on resting calcium. However, this is an important question for the future, Thanks for suggesting it.

  1. Although subtle changes (Fig1) were observed in the calcium current of bipolar cells still it is important to show if the calcium channels are miss-localized or how they look in terms of their localization with SR  

Response: This is one of our future goals to address. Due to the limitations in the number of old fish, we are unable to do those experiments at this time.

  1. Does calcium seem to be for critical of aging defects in the bipolar cells if not what other factors may be contributing towards physiological changes in the aging retina? what are authors insight on it.  

Response: Calcium dysregulation is a common factor altered in any age-related disease, for example, in Alzheimer’s disease in the brain and retina (1. Small, D.H. Dysregulation of Calcium Homeostasis in Alzheimer’s Disease. Neurochem Res 34, 1824–1829 (2009). https://doi.org/10.1007/s11064-009-9960-5; 2. Kiani, L. Calcium dysregulation could underlie lysosomal impairment in AD. Nat Rev Neurol 19, 65 (2023). https://doi.org/10.1038/s41582-022-00766-y; 3. Wang, Y., Shi, Y., & Wei, H. (2017). Calcium Dysregulation in Alzheimer's Disease: A Target for New Drug Development. Journal of Alzheimer's disease & Parkinsonism7(5), 374. https://doi.org/10.4172/2161-0460.1000374). The current paper is focused on normal aging. It is likely that the altered calcium we see in normal aging can alter the vision. We have added this explanation in the discussion now (lines 565-567).

Reviewer 2 Report

This is an excellent paper on a very important but understudied topic: the aging of ribbon type synapses in the inner plexiform layer (IPL) of the vertebrate retina. All visual information is first transmitted by ribbon synapses so the paper is very important for the field of mechanisms of visual processing. The paper is very well written and the data set is solid and well analyzed. Major new findings are clearly described with the use of top-notch techniques. The major findings are: 1) reduction in ribbon number and shape change with aging, 2) reduction in local Ca rises, which fits nicely with reduction in ribbon numbers, since Ca channels tend to cluster near ribbons, 3) reduction in the overall size of the IPL, suggesting a loss of synapses in the IPL and maybe even cell death in some types of amacrine and/or ganglion cells, or the retraction of dendrites from amacrine and ganglion cells, which will reduce the size of the IPL.

Major: 

1)    Please report in Methods the average plus/minus SEM for the series resistance and membrane resistance for the voltage-clamp Ca current recordings and membrane capacitance measurements. And the average resting membrane capacitance value? The pipette was placed in soma or terminal of the rod bipolar cell?

2)    Please report as an extra bar graph the exocytosis efficiency, which is just the ratio of Cm jump to Ca current charge (units of fF/pC). This number will be useful to compare with larval zebrafish cells and other ribbon synapses. How do the reported Ca current amplitude and Cm jumps in MA compare to previous larval fish recordings?

3)    In Figure 1 caption what was the N=? for MA cells?

4)    The number of ribbons changes from 38 in MA to 15 in OA. This is a 2-fold reduction in ribbon number and is quite impressive!! The 2-fold reduction should be mentioned in the Abstract.

5)    In zebrafish and goldfish, ribbons from bipolar cells and photoreceptors are known to disappear at night and reappear in the morning. This diurnal synaptic plasticity of ribbons with circadian rhythm may become impaired with age (please see and cite Hull et al., JNeurophysiol., 96: 2025–2033, 2006) and this putative impairment may thus cause the effects with aging.

6)    The results of a reduction in the overall width of IPL are very interesting and have also been seen in human retina with some types of retinal diseases (like Parkinsons). In this regard the authors should discuss a bit that a possible consequence of a loss of ribbons is that the postsynaptic processes or partners will also be lost. This putative retraction the processes has been seen in ribbon synapses of hair cells and the mechanism may be Ca-permeable AMPA receptors in the postsynaptic processes. The authors should discuss this issue as an extra paragraph in the Discussion on the mechanisms that may lead to IPL reduction in size and retraction of dendritic processes. Please see and cite for example Sebe et al. (J. Neurosci. 37: 6162–6175, 2017; for hair cell synapses with Ca-permeable AMPA receptors in zebra finch and frog) and Kim and von Gersdorff (Neuron 89, 507–520, 2016, for ribbon synapses with Ca-permeable AMPA receptors in goldfish retina).

7)    One implication of the lower Ca influx with age but the same whole-cell Ca current amplitude is that Ca channels may be less well localized near ribbons. So maybe the concentration of mobile Ca binding proteins is reduced in the rod bipolar cells with age to compensate for this and still allows normal exocytosis (Cm jumps). I like this idea that the authors expose since a reduction in Ca buffering proteins may cause a ribbon loss and eventual loss of bipolar cell terminals with age and size reduction in the IPL.

Author Response

Reviewer 2

This is an excellent paper on a very important but understudied topic: the aging of ribbon type synapses in the inner plexiform layer (IPL) of the vertebrate retina. All visual information is first transmitted by ribbon synapses so the paper is very important for the field of mechanisms of visual processing. The paper is very well written and the data set is solid and well analyzed. Major new findings are clearly described with the use of top-notch techniques. The major findings are: 1) reduction in ribbon number and shape change with aging, 2) reduction in local Ca rises, which fits nicely with reduction in ribbon numbers, since Ca channels tend to cluster near ribbons, 3) reduction in the overall size of the IPL, suggesting a loss of synapses in the IPL and maybe even cell death in some types of amacrine and/or ganglion cells, or the retraction of dendrites from amacrine and ganglion cells, which will reduce the size of the IPL.

Response: Thank you to this reviewer for their great interest and constructive feedback on our manuscript.

Major: 

1)    Please report in Methods the average plus/minus SEM for the series resistance and membrane resistance for the voltage-clamp Ca current recordings and membrane capacitance measurements. And the average resting membrane capacitance value? The pipette was placed in soma or terminal of the rod bipolar cell?

Response: The pipette was placed in the terminals. We have now provided the series resistance and membrane resistance for the voltage-clamp Ca current recordings and membrane capacitance measurements, as well as the average resting membrane capacitance value under methods (lines 130-133). Of note, we found that resting capacitance for OA zebrafish is higher than MA, which is consistent with our findings that the size of soma and terminals are larger in OA. We thank the reviewer for the great suggestion, and now we have a new find to report.

2)    Please report as an extra bar graph the exocytosis efficiency, which is just the ratio of Cm jump to Ca current charge (units of fF/pC). This number will be useful to compare with larval zebrafish cells and other ribbon synapses. How do the reported Ca current amplitude and Cm jumps in MA compare to previous larval fish recordings?

Response: Thank you for the great comments and suggestions. We have now provided this information as Supplemental Figure-1 for 0.2mM EGTA.

            We found the exocytosis efficiency in MA zebrafish for brief stimuli (0.98 ± 0.14) is lower compared to previous findings reported for hair ribbon synapses in larval zebrafish in response to longer/sustained stimuli, ~1.42. These differences could be due to differences in the age of the fish, pipette solution, calcium channel subunits, number of vesicles associated with synaptic ribbons, and strength of stimuli. We have discussed these points in the discussion (lines 485-490).

Sheets L, He XJ, Olt J, et al. Enlargement of Ribbons in Zebrafish Hair Cells Increases Calcium Currents But Disrupts Afferent Spontaneous Activity and Timing of Stimulus Onset. J Neurosci. 2017;37(26):6299-6313. doi:10.1523/JNEUROSCI.2878-16.2017

Lv C, Stewart WJ, Akanyeti O, et al. Synaptic Ribbons Require Ribeye for Electron Density, Proper Synaptic Localization, and Recruitment of Calcium Channels. Cell Rep. 2016;15(12):2784-2795. doi:10.1016/j.celrep.2016.05.045

3)    In Figure 1 caption what was the N=? for MA cells?

Response: We apologize for not providing the information we have now provided in the figure legends (line 227-229).

4)    The number of ribbons changes from 38 in MA to 15 in OA. This is a 2-fold reduction in ribbon number and is quite impressive!! The 2-fold reduction should be mentioned in the Abstract.

Response: Thank you for the great suggestion. We have provided this information in the abstract (line 33).

5)    In zebrafish and goldfish, ribbons from bipolar cells and photoreceptors are known to disappear at night and reappear in the morning. This diurnal synaptic plasticity of ribbons with circadian rhythm may become impaired with age (please see and cite Hull et al., JNeurophysiol., 96: 2025–2033, 2006), and this putative impairment may thus cause the effects with aging.

Response: Great point, and we will plan to address this question when we access more older fish. We have added this point as one of the future goals in the discussion (lines 552-556).

6)    The results of a reduction in the overall width of IPL are very interesting and have also been seen in human retina with some types of retinal diseases (like Parkinson's). In this regard, the authors should discuss a bit that a possible consequence of a loss of ribbons is that the postsynaptic processes or partners will also be lost. This putative retraction the processes has been seen in ribbon synapses of hair cells and the mechanism may be Ca-permeable AMPA receptors in the postsynaptic processes. The authors should discuss this issue as an extra paragraph in the Discussion on the mechanisms that may lead to IPL reduction in size and retraction of dendritic processes. Please see and cite, for example, Sebe et al. (J. Neurosci. 37: 6162–6175, 2017; for hair cell synapses with Ca-permeable AMPA receptors in zebra finch and frog) and Kim and von Gersdorff (Neuron 89, 507–520, 2016, for ribbon synapses with Ca-permeable AMPA receptors in goldfish retina).

Response: Thank you for the great suggestion. We have emphasized this information now in discussion. (please see line 557-563).

7)    One implication of the lower Ca influx with age but the same whole-cell Ca current amplitude is that Ca channels may be less well localized near ribbons. So maybe the concentration of mobile Ca binding proteins is reduced in the rod bipolar cells with age to compensate for this and still allows normal exocytosis (Cm jumps). I like this idea that the authors expose since a reduction in Ca buffering proteins may cause a ribbon loss and eventual loss of bipolar cell terminals with age and size reduction in the IPL.

Response: Thank you for the acknowledgment.

Reviewer 3 Report

This study of calcium dynamics at the aging bipolar cell ribbon synapse is truly novel and the questions that remain after reading the paper are said to be the subject of future study. I am familiar with the work of these authors and have personal experience studying similar topics with many of the same techniques.  All experience were performed with appropriate controls, number of cells, and the authors have pre-identified and addressed limitations of their study. I found no major flaws and recommend that paper to be accepted, after clarification of minor aspects.

This paper by Shrestha et al. examines how senescence changes calcium-dependent exocytosis at bipolar cell ribbon synapses of zebrafish retina. Voltage-clamp of isolated bipolar cells and membrane capacitance measurements were used to assess calcium currents and vesicle exocytosis, respectively. Measuring membrane capacitance is technologically difficult, but an excellent technique for nearly direct measurement of exocytosis. They further assessed age-related morphological changes by counting and measuring synaptic ribbons, and visualizing the inner plexiform layer organization by immunofluorescence. It is currently believed that the apposition of the ribbon near and above voltage-gated calcium channels is responsible for the creation of the calcium nano-domains that allow for exocytosis of docked vesicles. Thus, the authors looked at calcium dynamics by imaging with a fluorescent calcium indicator under different intracellular buffering conditions.

All techniques were sound and appropriate for making the conclusions that the authors do, recognizing that not all labs have access to TIRF microscopy. At first, I was skeptical about the resolution of confocal for Ca2+ imaging, compared to 2-photon microscopy, however this has recently been directly compared, and the signal-to-noise ratio is better with confocal. I have a few questions that can be addressed in the paper or in reply directly to me, if it is something that is widely known by others.

1.       As for resolution of confocal microscopy, is it truly resolute enough in live cell imaging to measure the size of synaptic ribbons or distinguish between two that are near to one another? What is the error expected? The literature search I did yielded studies that measured and counted with fixed tissues under fluorescence or electron microscopy.

2.       The paper mentions the loss of the large transient, seen by Ca2+ imaging, but the kinetics found by voltage-clamp were not reported. Could this help identify whether there has been a shift of proportions of L-type to T-type calcium channels?

3. The authors conclude that despite similar calcium currents and exocytosis in mature and old age bipolar cells, the change in local calcium dynamics is due to the shift from many smaller ribbons to a smaller number of larger ribbons.  Zenisek’s 2004 paper shows that RIBEYE, the protein the authors used to mark ribbons for measurement, creates a diffusion barrier that promotes the formation of sharp Ca2+ nanodomains near channels right below the synaptic ribbon. It seems 

4. In the legend for Figure 4 C-D, there is a parenthetical that says "provide calcium currents, " if this wasn't an accidental omission, could the meaning of this be clarified?

Author Response

This study of calcium dynamics at the aging bipolar cell ribbon synapse is truly novel and the questions that remain after reading the paper are said to be the subject of future study. I am familiar with the work of these authors and have personal experience studying similar topics with many of the same techniques.  All experience were performed with appropriate controls, number of cells, and the authors have pre-identified and addressed limitations of their study. I found no major flaws and recommend that paper to be accepted, after clarification of minor aspects.

This paper by Shrestha et al. examines how senescence changes calcium-dependent exocytosis at bipolar cell ribbon synapses of zebrafish retina. Voltage-clamp of isolated bipolar cells and membrane capacitance measurements were used to assess calcium currents and vesicle exocytosis, respectively. Measuring membrane capacitance is technologically difficult, but an excellent technique for nearly direct measurement of exocytosis. They further assessed age-related morphological changes by counting and measuring synaptic ribbons, and visualizing the inner plexiform layer organization by immunofluorescence. It is currently believed that the apposition of the ribbon near and above voltage-gated calcium channels is responsible for the creation of the calcium nano-domains that allow for exocytosis of docked vesicles. Thus, the authors looked at calcium dynamics by imaging with a fluorescent calcium indicator under different intracellular buffering conditions.

All techniques were sound and appropriate for making the conclusions that the authors do, recognizing that not all labs have access to TIRF microscopy. At first, I was skeptical about the resolution of confocal for Ca2+ imaging, compared to 2-photon microscopy, however this has recently been directly compared, and the signal-to-noise ratio is better with confocal. I have a few questions that can be addressed in the paper or in reply directly to me, if it is something that is widely known by others.

Response: Thank you to this reviewer for their great interest and constructive feedback on our manuscript.

  1. As for resolution of confocal microscopy, is it truly resolute enough in live cell imaging to measure the size of synaptic ribbons or distinguish between two that are near to one another? What is the error expected? The literature search I did yielded studies that measured and counted with fixed tissues under fluorescence or electron microscopy.

Response: Thank you for the reviewer’s observation and for appreciating the limits of the confocal microscope. From a theoretical standpoint, the resolution in fixed tissue should be no better than with live cell imaging, but we acknowledge that the use of peptide label in live cell rather than antibodies used in traditional immunohistochemistry may affect resolution. Per the reviewer’s concern, we performed new experiments with fixed tissues. However, we were only able to conduct such measurements for MA as OA fish are currently not available in our vivarium and could not get from the vendors (ZIRC). Please note that the ribbon counts in MA fish from fixed tissues were identical to those obtained with live imaging (38.2 ± 0.5; n=5 cells in 3 animals).

Also, the point-spread function of the microscope, determined by fitting a 2D Gaussian to images of single 27-nm fluorescent beads, shows the x-width was 268 nm in the lateral, and the y-width was 273 nm resolution. Considering the ribbon size measured by us with electron microscopy for zebrafish retinal bipolar cells ~250 nm (please see Figure -5 in Vaithianathan T, Zanazzi G, Henry D, Akmentin W, Matthews G. Stabilization of spontaneous neurotransmitter release at ribbon synapses by ribbon-specific subtypes of complexin. J Neurosci. 2013;33(19):8216-8226. doi:10.1523/JNEUROSCI.1280-12.2013) we believe the resolution of confocal microscopy is sufficient to distinguish between two ribbons in most cases, though we cannot rule out that in some cases two closely localized ribbons will be indistinguishable. As for measurements of size, confocal microscopy is insufficient to measure precise size of ribbons, but an increase in the size of sufficient magnitude will result in a broadening of the individual ribbon spot, which is detectable as we see here.  

We have added a new author, Dr. Courtney E. Frederick, to acknowledge her contribution to the new data obtained to address the synaptic ribbon numbers in fixed tissues. 

  1. The paper mentions the loss of the large transient, seen by Ca2+ imaging, but the kinetics found by voltage-clamp were not reported. Could this help identify whether there has been a shift of proportions of L-type to T-type calcium channels?

Response: We thank the reviewer for this great suggestion. The terminals of Mb1 bipolar cells primarily have L-type calcium channels (Heidelberger R, Matthews G. Calcium influx and calcium current in single synaptic terminals of goldfish retinal bipolar neurons. J Physiol. 1992;447:235-256. doi:10.1113/jphysiol.1992.sp019000) However, it is possible that during aging changes in calcium channel subtypes could happen. Of note, as shown in Fig. 1 A and B, we did not find any changes in the kinetics of the calcium current as would be expected from contribution of T-type channels. Thus, we don’t anticipate changes in the kinetics of calcium current. We suspect the absence of a large transient could be due to calcium channels not being as well localized to synaptic ribbons, but further experiments would be needed to confirm this hypothesis.

  1. The authors conclude that despite similar calcium currents and exocytosis in mature and old age bipolar cells, the change in local calcium dynamics is due to the shift from many smaller ribbons to a smaller number of larger ribbons.  Zenisek’s 2004 paper shows that RIBEYE, the protein the authors used to mark ribbons for measurement, creates a diffusion barrier that promotes the formation of sharp Ca2+ nanodomains near channels right below the synaptic ribbon. It seems 

Response: We thank the reviewer for this comment. Zenisek’s 2004 paper established that calcium entry sites colocalized with Ribeye, but not ribbon act as a barrier. The hotspots that denoted the calcium entry sites were most visible in the presence of high concentrations of EGTA. Under these conditions, calcium entering the cell is preferentially visualized due to the mismatch of kinetics between the fast-binding calcium indicator and the slow-binding EGTA. While a ribbon barrier might enhance this signal, hotspots would also be expected to be visible even without it.  We could not fully address the reviewer’s comments entirely since the sentence ended in the middle.

  1. In the legend for Figure 4 C-D, there is a parenthetical that says "provide calcium currents, " if this wasn't an accidental omission, could the meaning of this be clarified?

Response: Sorry for the typo-we have provided this information in the revised version (lines 362-363).

Reviewer 4 Report

While it is certainly intriguing, the majority of these observations lack robust underlying mechanisms.

 In Figure 1, particularly in 1D, although no significant differences are apparent, there is a conspicuous trend indicating higher levels in OA compared to MA. To bolster the data's reliability, the authors should augment the number of biological replicates and consider utilizing dot plots to depict individual biological replicates alongside bar charts.

 To enhance the data's persuasiveness and credibility, the authors should explicitly specify the number of individual zebrafish and cells included in each dataset. Additionally, they should clearly delineate the statistical methods employed in their analysis.

Author Response

While it is certainly intriguing, the majority of these observations lack robust underlying mechanisms.

 In Figure 1, particularly in 1D, although no significant differences are apparent, there is a conspicuous trend indicating higher levels in OA compared to MA. To bolster the data's reliability, the authors should augment the number of biological replicates and consider utilizing dot plots to depict individual biological replicates alongside bar charts.

Response: Thank you for the reviewer’s suggestion. We have now provided this information in Supplemental Figures C and D. Unfortunately, the availability of OA animals is limiting in these experiments.

 To enhance the data's persuasiveness and credibility, the authors should explicitly specify the number of individual zebrafish and cells included in each dataset. Additionally, they should clearly delineate the statistical methods employed in their analysis.

Response: Thank you for the reviewer’s suggestion. We have provided animal and cell numbers for each figure as below:

Figure 1: Please check lines 227-229.

Figure 2: Please check lines 267-268

Figure 3: Please check lines 309-311.

Figure 4: Please check lines 362-365.

Figure 5: Please check lines 372-376.

Statistical methods: Variance in estimates of the population mean is reported as ± sem. Statistical significance of differences in average amplitudes of calcium current, capacitance, synaptic ribbons size and number, calcium transients rare were assessed using unpaired, two-tailed t-tests with unequal variance. We have now provided this information under methods. Please check lines 189-196.  

Round 2

Reviewer 4 Report

The authors have addressed all my concerns and the manuscript is at better shape now.